# Real-World Data of First 12-Months of Ofatumumab Treatment in Multiple Sclerosis Patients—A Multicenter Experience from Tertiary Referral Centers

**DOI:** 10.3390/medicina61091568

**Published:** 2025-08-31

**Authors:** Weronika Galus, Aleksandra Kaczmarczyk, Anna Walawska-Hrycek, Joanna Siuda, Milena Polewka, Anetta Lasek-Bal, Przemysław Puz

**Affiliations:** 1Department of Neurology, Faculty of Medical Sciences in Katowice, Medical University of Silesia, 40-752 Katowice, Poland; akaczmarczyk@sum.edu.pl (A.K.); awalawska-hrycek@sum.edu.pl (A.W.-H.); jsiuda@sum.edu.pl (J.S.); 2Students’ Scientific Association, Department of Neurology, Faculty of Health Sciences in Katowice, Medical University of Silesia, 40-055 Katowice, Poland; s86110@365.sum.edu.pl; 3Department of Neurology, Faculty of Health Sciences in Katowice, Medical University of Silesia, 40-635 Katowice, Poland; alasek-bal@sum.edu.pl (A.L.-B.); ppuz@sum.edu.pl (P.P.)

**Keywords:** multiple sclerosis, disease modifying therapy, highly effective therapy, ofatumumab, real-world data

## Abstract

*Background and Objectives:* Ofatumumab (OFA) is the first fully human anti-CD20 monoclonal antibody approved for the treatment of RRMS classified as a high-efficacy treatment agent. Real-world evidence is essential for evaluating the effectiveness and safety of OFA. *Materials and Methods*: A total of 184 patients (72.3% women, mean age 38 years (±10.9), 51 naïve patients and 133 after switch. Among them, 142 patients were evaluated after first 12-months of treatment according to relapse rate, neurological status expressed by Expanded Disability Status Scale (EDSS), new T2-weighted (T2-w) lesions and gadolinium-enhancing lesions (GELs) in magnetic resonance imaging (MRI), confirmed disability progression, as well as adverse events. Logistic regression identified factors associated with disease activity at ofatumumab initiation, including age, sex, disease duration, prior treatment, and baseline EDSS. *Results*: After the first 12 months of OFA treatment, relapses occurred in 12.0% of patients; new or enlarging T2-w lesions were observed in 12.7%; GELs in 3.5%; EDSS progression in 12.7%; and EDSS improvement in 14.2%. The No Evidence of Disease Activity-3 (NEDA-3) status was achieved in 76.1% of patients overall—75.8% in those who switched from another disease-modifying therapy (DMT), and 76.6% in treatment-naïve individuals. No significant differences were observed between the naïve and switch groups. Baseline EDSS at ofatumumab initiation was a significant predictor of relapse activity, while age was significantly associated with MRI activity (GELs) at 12 months. *Conclusions*: Real-world data confirmed high efficacy and safety of ofatumumab in RRMS. NEDA-3 was achieved more often than in registration trials. No efficacy differences between naïve and switch patients were observed.

## 1. Introduction

Multiple sclerosis (MS) is an autoimmune disease of the central nervous system (CNS) predominantly develops between the ages of 20 and 40, with an increasing global prevalence [1], including Poland [2]. Neuroinflammation leads to demyelination, neurodegeneration and axonal loss responsible for disability progression [3]. MS is still an incurable disease, but current disease-modifying therapies (DMTs) prolong MS patients’ quality lifetime with no clinical and radiological disease progression [4]. Generally, the efficacy of treatment is assessed in relation to the No Evidence of Disease Activity 3 (NEDA-3) parameter defined as the absence of relapses, stable neurological condition evaluated by the Expanded Disability Status Scale (EDSS) [5], and no new/enlarging hyperintense lesions on T2-weighted (T2-w) or gadolinium enhancing lesions (GELs) in T1-weighted (T1-w) on magnetic resonance imaging (MRI) [6]. Additionally, MS clinical trials employ various methods to monitor disease progression, including Confirmed Disability Progression (CDP) evaluating specific increase in EDSS [7] or brain atrophy progression [8].

The role of lymphocytes B in the pathogenesis of MS has recently been highlighted largely due to discovery of high-efficacy treatment agents (HETA) with anti-CD20 monoclonal antibodies (mAb) that cause lymphocytes B depletion [9]. The first anti-CD20 mAb that showed efficacy in suppression of inflammation in relapsing-remitting multiple sclerosis (RRMS) was rituximab–a murine–human chimeric mAb commonly used for off-label treatment of MS [10]. This was followed by the development of another anti-CD20 mAbs such as ocrelizumab–humanized anti-CD20 mAb [11] and ublituximab–chimeric antiCD20 mAb [12].

Ofatumumab (OFA) is the first fully human anti-CD20 mAb approved for the treatment of RRMS [13] that binds to a distinct CD20 epitope and cause almost complete lymphocyte B depletion in a complement-dependent cytotoxicity (CDC) mechanism at a lower concentration than others anti-CD20 antibodies [14]. In the first clinical investigation of OFA in MS, a randomised placebo-controlled (RCT) phase 2 study, a >99% reduction in gadolinium-enhancing T1 lesions was recorded as well as lymphocyte B depletion [15]. Subsequently, the phase 2b placebo-controlled MIRROR study confirmed the >90% lesion reduction and enable the estimation of safety of subcutaneous (SC) doses supported by model studies, that significantly decrease the number of infusion-related reactions (IRRs) [16]. In ASCLEPIOS I and II phase 3 (ASCLEPIOS I/II) trials, OFA SC, compared with teriflunomide, was associated with significantly lower annual relapse rates (ARRs), reduced risk of 3- and 6-month disability worsening and nearly complete reduction of GELs. Adverse events (AEs) rates were similar with teriflunomide treatment, including infections and malignancies [13]. OFA can be self-administered by the patient at home SC every month without premedication with an initial dose of 20 mg (in 0.4 mL) at 0, 1 and 2 weeks [17]. In ASCLEPIOS I/II, the SC formulation of OFA was associated with high levels of compliance (98.8%). OFA injections have been mostly well-tolerated, the occurrence of local and systemic IRRs including headache, fever and influenza-like symptoms was highest after the first injection and decreased noticeably with following injections [18].

Based on the results of the RCTs, the FDA registered OFA for the treatment of patients with RRMS in August 2020, and the EMA in March 2021. As of 1 November 2022, OFA has been approved for the treatment of patients with RRMS in the Therapeutic Program for the treatment of MS of the Ministry of Health in Poland (No, B.29) [19].

## 2. Materials and Methods

### 2.1. Study Participants

This is a retrospective cohort study. Among patients with RRMS treated in Polish Ministry of Health Multiple Sclerosis Therapeutic Program in following medical centers in Upper Silesia, Poland: University Medical Centre of Medical University of Silesia, Katowice, Poland and Upper Silesia Medical Centre of Medical University of Silesia, Katowice, Poland; authors selected 184 consecutive patients who started OFA therapy from December 2022 till December 2024. All patients were adults (>18 years old) and diagnosed with RRMS according to McDonalds criteria from 2010 [20] and 2017 [21].

### 2.2. Patient Eligibility for Ofatumumab Therapy

Patients were enrolled according to the eligibility rules of the Polish Ministry of Health Multiple Sclerosis Therapeutic Program Polish National (B.29). Inclusion criteria for OFA comprised: diagnosis of RRMS based on the McDonald criteria confirmed by MRI, evidence of disease activity within the last 12 months (at least one clinical relapse or one GELs), an EDSS score between 0 and 4.5, and age ≥18 years, with no contraindications listed in the Summary of Product Characteristics. The program also allows treatment continuation for patients previously receiving OFA under other reimbursement schemes, provided they met the above requirements. Furthermore, patients could be switched to OFA from other I-line therapies in case of adverse events, lack of efficacy (≥1 relapse, ≥1 GEL, or ≥2 new T2-w lesions), or when a therapeutic benefit was expected by the treating physician. De-escalation from higher-efficacy therapies (fingolimod, natalizumab, alemtuzumab, ocrelizumab, cladribine) to OFA was also permitted for safety reasons.

### 2.3. Evaluation of the Efficacy and Safety of Ofatumumab Treatment

Patients were evaluated clinically and radiologically in accordance with the requirements of the Therapeutic Program in Multiple Sclerosis. Clinical assessments included neurological examination with EDSS scoring, review of medical history, and routine laboratory tests (complete blood count, liver and renal function tests, urinalysis, and pregnancy test when indicated), performed every 3 months. Radiological monitoring consisted of annual contrast-enhanced brain MRI performed every 12 months. Data from medical documentation and the Therapeutic Program Monitoring System (System Monitorowania Programów Terapeutycznych, SMPT) were used to collect information on relapses (number and severity), neurological status (EDSS), MRI findings (new or enlarging T2-w lesions and gadolinium-enhancing lesions), as well as adverse events and infections. Moreover, in the group after switch from other DMT, the reason for changing treatment to OFA was collected.

EDSS was assessed by board-certified neurologists experienced in MS at each site, using a harmonized scoring checklist based on the Kurtzke scale [5]. Where feasible, the same clinician assessed the same patient longitudinally to reduce inter-rater variability. Additionally, the disease progression was evaluated by EDSS change defined as min. 1.5-point increase for a baseline EDSS of score 0, a 1-point increase for a baseline EDSS between 1 and 5 and 0.5-point increase for a baseline EDSS of 5.5 or higher, sustained for two or more consecutive visits separated by >90 days. Relapse was defined as a clinical episode lasting at least 24 h, occurring in the absence of fever, infection, or other acute illness, and preceded by a minimum of 30 days without clinical relapse.

MRI activity was confirmed by the presence of new or enlarged T2-w lesions and/or GELs. All patients underwent brain MRI with a protocol complying with the Recommendations of the Polish Medical Society of Radiology and the Polish Society of Neurology for routinely used MRI in patients with multiple sclerosis [22].

NEDA-3 status was assigned to patients without relapses, EDSS progression, or MRI activity.

The 12-month effectiveness analysis included only patients who had completed the annual visit by the database-lock date. Patients who had not yet completed 12 months of exposure at database lock were not included in this analysis and remain under follow-up.

Data on adverse events and infections were collected primarily from patients’ medical documentation and the Therapeutic Program Monitoring System. When information on the clinical course or severity grading was available, it was extracted and classified accordingly. Infections were grouped into respiratory, urinary-tract, mucocutaneous viral (HSV), and gastrointestinal categories based on clinical documentation. Severity was graded using the Common Terminology Criteria for Adverse Events (CTCAE) v5.0 (Grade 1–5) [23]. Where available, we recorded route of anti-infective therapy (none/topical/oral/intravenous) and outcomes (resolved/resolving/ongoing).

The following analyses were conducted: a comparison of treatment efficacy and safety between treatment-naïve patients (receiving OFA as first-line DMT) and those switched from other DMTs, as well as a multivariate assessment of the impact of selected factors on MS disease activity, including relapses, CDP, and MRI findings.

### 2.4. Statistical Analysis

Statistical analysis was performed using STATISTICA 13.0 PL (Tibco Software Inc., Palo Albo, CA, USA), and R software (version 4.4.0; R Foundation for Statistical Computing, Vienna, Austria).

A significance level of α = 0.05 was applied to control the Type I error rate. Descriptive statistics were used to summarize the data: continuous variables were presented as mean ± standard deviation or median, as appropriate, while categorical variables were presented as counts or frequencies (n) and percentages (%).

The *t*-test of Wilcoxon rank-sum test compared continuous variables between two independent groups, while Pearson’s chi-square test or Fisher’s exact test was used for categorical variables based on sample size.

Logistic regression with a logit link function analyzed the effect of multiple variables (age, sex, disease duration, therapy initiation, EDSS at treatment initiation) on binary outcomes (relapse, MRI activity, EDSS progression, NEDA-3). Confidence intervals and *p*-values were calculated using the asymptotic z-test approximation.

Logistic regression with a logit link function was used to assess the effect of multiple variables—including age, sex, disease duration, type of therapy initiation (naïve vs. switch), and baseline EDSS)—on binary outcomes: relapse occurrence, MRI activity, EDSS progression, and achievement of NEDA-3. Confidence intervals and *p*-values were estimated using the asymptotic z-test approximation.

For all proportions, 95% confidence intervals (95% CI) were calculated using the Wilson score method; odds ratios from logistic regression are presented with model-based 95% CIs.

## 3. Results

### 3.1. Study Group Characteristics

At the baseline, there were 184 (133 women and 51 men, mean age was 37.96 ± 10.95 years) patients treated with OFA; 133 patients after switch from another DMT and 51 naïve patients. Of the 184 patients treated with OFA, 142 completed 12 months of observation with a full clinical and radiological assessment (47 naïve patients and 95 patients after switch). Naïve patients were statistically younger and had shorter disease duration than patients after switch (Table 1). 42 patients had not yet reached the annual assessment at database lock; 40 of them were still on treatment, while 2 had discontinued therapy as detailed in Section 3.4. The study flow is illustrated in Figure 1.

Prior treatment regimens included various DMTs, such as dimethyl fumarate (51 patients), teriflunomide (14), interferon beta (10), glatiramer acetate (9), ozanimod (4), fingolimod (2), natalizumab (2), cladribine (2), and alemtuzumab (1), before switching to ofatumumab.

### 3.2. Clinical and Radiological Efficacy of Ofatumumab Treatment After First 12-Months

Within the first 12-months of treatment, clinical relapses were documented in 17 patients (12%), new or enlarged T2-w lesions in 18 patients (12.67%), and GELs in 5 patients (3.52%). EDSS progression was reported in 18 patients (12.67%), EDSS improvement in 20 patients (14.18%). NEDA-3 parameter was achieved in 108 patients (76.06%) during the observation period. There were no statistically significant differences in the clinical and radiological disease activity parameters during OFA treatment between naïve and patients after switch (Table 2). Exact 95% CIs for all point estimates are reported in Appendix A.

### 3.3. Adverse Events

Within the first year of treatment, 123 patients (66.9%) experienced at least one AE. The majority of AE were mild, and no serious adverse events (SAEs) were reported. The most common AE were IRRs, reported in 116 patients (63.04%), primarily following the initial doses (first and second doses). A total of 30 patients (16.3%) experienced infections during the treatment period. Overall, 63 infections were reported. Respiratory tract infections accounted for 34 cases, urinary tract infections for 14 cases, herpes simplex virus (HSV) infections for 11 cases, and gastrointestinal tract infections for 4 cases. All infections were graded as 1–2 according to the CTCAE; there were no Grade ≥ 3 infections, no hospitalizations, and no SAEs. Exact 95% CIs for AE proportions are provided in Appendix A. There were no statistically significant differences in AE between naïve and switched patients after 12-months of OFA treatment (Table 3).

### 3.4. Ofatumumab Discontinuation

Among the study group, two of patients discontinued OFA treatment during the observation period. One patient discontinued due to chronic urinary tract infections and ongoing disease activity, while the other discontinued due to disease activity alone. Both discontinuations occurred before 12 months of exposure; therefore, these patients were excluded from the 12-month effectiveness analysis.

### 3.5. Factors Determining Clinical and Radiological Activity of the Disease

A logistic regression analysis was performed to identify factors associated with clinical and radiological disease activity at the initiation of OFA treatment, including age, sex, disease duration, prior treatment, and baseline EDSS score. Baseline EDSS at the initiation of OFA treatment was a significant predictor of relapse activity, while age was significantly associated with MRI activity (presence of GELs) after one year of treatment (Table 4).

### 3.6. Reasons in Switching Therapy into Ofatumumab

The main reason for switching from prior therapy to OFA were relapse activity (44 patients, 46.3%), radiological activity (45 patients, 47.4%), and adverse events of previous therapy (30 patients, 32.6%). Additionally, 28 patients (29.5%) were switched to OFA due to clinicians’ decision expected therapeutic benefits. Other reasons for switching included EDSS progression (3 patients, 3.2%), de-escalation therapy (4 patients, 4.2%) and DMT reinitiation after pregnancy and childbirth (1 patient). A total of 64 patients had previously been treated with more than one DMT.

## 4. Discussion

This study presents real-world data on OFA treatment in patients with RRMS, reflecting clinical practice conditions distinct from those of RCT. The main findings include the high effectiveness of OFA during the first year of therapy—NEDA-3 was achieved in 76.1% of patients—and the absence of significant differences in treatment outcomes between treatment-naïve patients and those switched from other DMTs.

In the ASCLEPIOS I/II registration studies, NEDA-3 was achieved in 48% of patients treated with OFA during the first 12 months of therapy [13], increasing to 80%, 87.5%, 89.2% and 93.4%, respectively, in subsequent years during the ALITHIOS extension study [24]. The value of NEDA-3 after the first 12 months of ofatumumab therapy of patients presented in our study (76.06%) is higher than the NEDA-3 obtained in OFA registration trials. These discrepancies may be attributed to methodological differences between our study and the registration trials. Our analysis included a real-world cohort of patients treated with OFA according to the criteria of Ministry of Health Multiple Sclerosis Therapeutic Program. In contrast, registration studies were phase III RCTs that enrolled different MS patient populations, often with higher baseline disease activity and stricter inclusion criteria. Moreover, the rigorous and standardized monitoring protocols applied in RCTs may have contributed to a higher detection rate of disease activity during the first year of treatment. Beyond study design, several factors may explain our higher 12-month NEDA-3 proportion. At baseline the cohort had low disability (median EDSS = 2.0), and treatment-naïve patients were younger with shorter disease duration—characteristics that favor short-term disease control. Treatment persistence was high (2/184 discontinuations), aligning with real-world observations that monthly self-administered SC OFA facilitates adherence and patient acceptability. In addition, a subset of switches occurred for reasons other than breakthrough disease (e.g., adverse events or anticipated benefit), potentially enriching for patients likely to stabilize on the new regimen. Together with the rapid B-cell depletion reported for OFA, these features may contribute to higher NEDA-3 in routine practice.

Our study also demonstrated low clinical and radiological disease activity during the first 12 months of OFA therapy, consistent with findings from other published RWE studies. In a 12-month study involving 175 patients, a significant reduction in relapse rates was observed, with only one relapse reported after approximately four months of treatment. MRI assessments demonstrated a decrease in new T2-weighted and gadolinium-enhancing lesions, while 90% of patients remained on therapy, indicating high treatment persistence [25]. Similarly, a German prospective study involving 81 patients with a mean observation period of 10 months reported only four relapses in the entire cohort and no evidence of EDSS progression [26]. Our results are in line with the recently published Polish multicenter study by Stępień et al., where 72.7% of 430 patients achieved NEDA-3 at 12 months, a proportion comparable to our cohort (76.1%), despite differences in baseline disability thresholds and broader inclusion criteria [27]. Similarly, Tai et al., in a U.S. claims-based analysis of 779 patients, demonstrated a 75% reduction in annualized relapse rate and fewer hospitalizations following ofatumumab initiation, further confirming its effectiveness in routine practice [28]. In addition, Zanghì et al. reported very high 12-month NEDA-3 rates (94.4%) and a low ARR (0.038) in an Italian real-world comparative cohort, with a favorable safety profile, supporting the robustness of ofatumumab efficacy across diverse healthcare [29]. Data from the ALITHIOS open-label extension study demonstrated sustained efficacy of OFA over five years, with consistently low annualized relapse rates (<0.05) and 93.4% of patients achieving no evidence of disease activity (NEDA-3) at year five [25,30]. A summary of published real-world studies on ofatumumab is presented in Table 5.

Our findings are consistent with other RWE and align with results from clinical trials, confirming the robust anti-relapse efficacy of OFA. Additionally, RWE suggests stability or improvement in disability status over time. Overall, observational data indicate that OFA substantially reduces relapse activity and effectively limits short-term disease progression under routine clinical conditions

We observed no significant differences in treatment efficacy between treatment-naïve patients and patients switched from other DMTs. Naïve patients met inclusion criteria based on clinical or radiological disease activity within the previous year. In the switched group, therapy was changed due to disease activity, intolerance to previous treatment, or the clinician’s decision based on expected therapeutic benefit. Despite differing reasons for initiating OFA therapy, both groups demonstrated comparable levels of disease control and stability over the 12-month observation period, confirming the high efficacy of OFA regardless of prior treatment status.

Our study also demonstrated a favorable safety profile of OFA. AEs were reported in 66.85% of patients initiating therapy, with the overall safety profile consistent with that observed in RCT. The most frequently reported adverse events in our cohort were IRRs and infections. According to the U.S. Food and Drug Administration (FDA) Adverse Event Reporting System, other commonly reported adverse events include fatigue, headache, chills, pyrexia, pain, nausea, vomiting, and infections such as nasopharyngitis, urinary tract infections, and pneumonia [31]. A recent FAERS-based pharmacovigilance analysis further confirmed these known risks but also identified additional potential off-label events, including brain fog, muscle spasms, and mood alterations, most frequently occurring during the first month of treatment [32]. Safety profiles presented in other RWE data align with RCTs data [25,26,27,28,29].

In our cohort, only two patients discontinued therapy within the 12-month period, indicating high adherence and persistence to OFA. These findings are consistent with other RWE, which generally demonstrate high adherence to OFA. The once-monthly subcutaneous administration and favorable tolerability profile likely contribute to sustained treatment persistence [25,33].

Our study also investigated the influence of selected baseline factors on disease activity and progression during OFA treatment, including age, sex, disease duration, prior use of DMTs, baseline EDSS score. In the multivariate logistic regression analysis, baseline EDSS was identified as a significant predictor of relapse occurrence, while age was significantly associated with MRI activity, specifically the presence of GELs, after 12 months of treatment. No other variables showed statistically significant associations with clinical or radiological disease activity, or with disability progression. The absence of additional significant predictors may be due to the relatively low number of confirmed events and the limited duration of follow-up. It is also possible that other unmeasured confounding factors contributed to disease activity in this cohort.

From a clinical point of view, our results allow us to confirm the high safety and efficacy of the OFA therapy in the RWE study based on a comparison of naive and switched patients, which can facilitate therapeutic decisions when choosing DMTs at any stage of the disease in patients with RRMS.

### 4.1. Strengths of the Study

Our findings are consistent with available RWE on the efficacy of OFA in treating patients with RRMS, both in treatment-naïve individuals and in those switched from other DMTs for various reasons. As an observational cohort study based on real-life clinical practice, our work contributes valuable insights into both the effectiveness and safety of OFA. RWE is essential for clinicians, as it reflects treatment outcomes under routine care conditions, which may differ from those reported in randomized controlled trials. By including data on disease activity and safety across different patient subgroups, our study supports more informed, individualized therapeutic decision-making. Furthermore, RWE can serve as a basis for population-level analyses, including cost-effectiveness assessments of MS therapies.

### 4.2. Limitations of the Study

The present study has several limitations that need to be considered when interpreting the results. First, the retrospective design may have introduced selection bias, as patients treated in tertiary referral centers may not be fully representative of the broader MS population. Second, there is a risk of information bias, as relapse events, MRI activity, or mild adverse events could have been under-reported or inconsistently documented in routine practice compared with the rigorous monitoring protocols of randomized controlled trials. Third, detection bias may have occurred due to variability in MRI protocols and EDSS scoring across participating centers. Because EDSS ratings were obtained under routine clinical care and we did not compute formal inter-rater reliability metrics across centers, some measurement variability cannot be excluded. This risk was mitigated by harmonized scoring guidance, same-rater follow-up whenever feasible, and the use of CDP confirmation. Moreover, despite multivariable adjustment, residual confounding (e.g., by unmeasured factors such as baseline MRI burden beyond the reported metrics, comorbidities, or lifestyle) cannot be excluded. Additionally, some confidence intervals were wide, reflecting limited precision, yet the estimates provide useful insights that warrant confirmation in larger studies that wide confidence intervals should be interpreted with caution and reflect the underlying data limitations. Finally, the relatively short follow-up period and the low number of clinical and radiological events limited the ability to identify robust predictors of disease activity or progression. Future prospective multicenter studies with longer observation and standardized monitoring will be necessary to validate our findings and to provide a more comprehensive evaluation of the long-term effectiveness and safety of OFA in real-world settings.

Another limitation of our study is the broader eligibility rules of the Polish Ministry of Health Multiple Sclerosis Therapeutic Program (B.29), which differ from the more restrictive inclusion and monitoring protocols of pivotal registration trials. As outlined in the Methods, this may introduce selection bias and limit direct comparability with RCTs, but it also reflects real-world practice and provides complementary evidence on the effectiveness and safety of OFA.

Generalizability of our findings applies mainly to healthcare systems that, like the Polish Therapeutic Program, enable early access to high-efficacy therapy, regular monitoring, and home subcutaneous administration. Outcomes may differ in settings with more limited access, less frequent monitoring, or less structured MS care.

## 5. Conclusions

Real-world data confirmed the high efficacy and favorable safety profile of OFA in patients with relapsing-remitting multiple sclerosis (RRMS). A higher proportion of patients achieved NEDA-3 status after 12 months of treatment compared to the pivotal registration trials. No statistically significant differences in treatment efficacy were observed between treatment-naïve patients and those who switched from previous therapies.

## Figures and Tables

**Figure 1 medicina-61-01568-f001:**
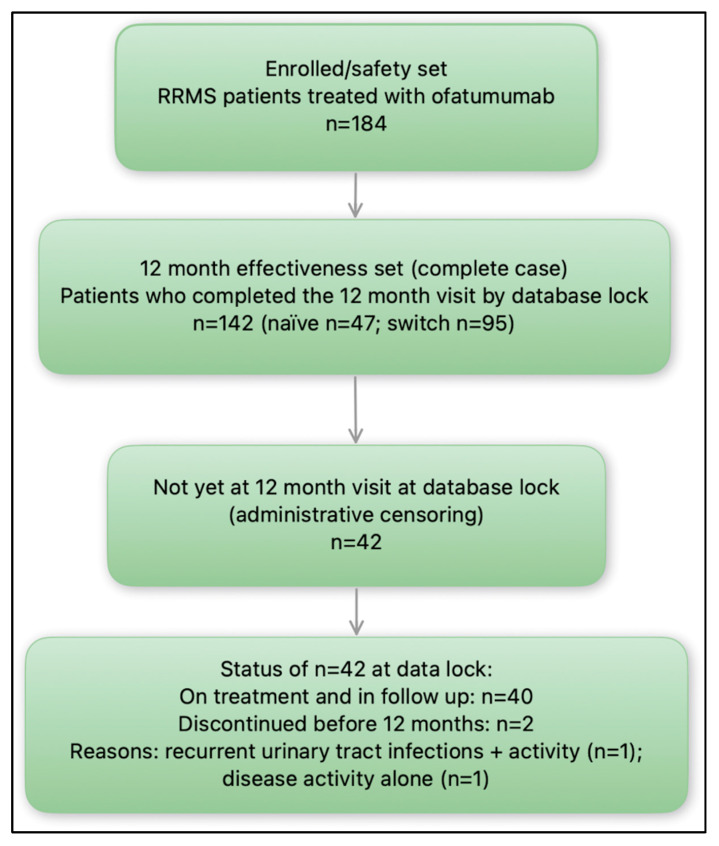
Flowchart of the study.

**Table 1 medicina-61-01568-t001:** Baseline characteristics of patients who completed 12-months of ofatumumab treatment, from both switch and naïve groups.

Characteristics	Patients After Switch form Another DMT (N = 95)	Naïve Patients (N = 47)	*p* *
Male/female [n, %]	22/73 (23.2/76.8)	14/33 (29.8/70.2)	0.39
Age [years ± SD]	39.47 ± 11.72	33.91 ± 8.71	0.29
Duration of disease [years ± SD]	11.11 ± 7.02	2.85 ± 1.48	<0.001
EDSS, [median, min–max)	2 (0–5.5)	2 (0–4.5)	0.07

DMT—disease modifying therapy, SD—standard deviation, EDSS—Expanded Disability Status Scale, * chi2 test.

**Table 2 medicina-61-01568-t002:** Clinical and radiological efficacy parameters after first 12-months of ofatumumab treatment between switch and naïve patients.

Efficacy of Treatment Parameters	Patients After Switch form Another DMT (N = 95)	Naïve Patients (N = 47)	*p* *
Relapses (n, %)	12 (12.6)	5 (10.6)	0.72
T2-w lesions (n, %)	12 (12.6)	6 (12.8)	0.35
GELs (n, %)	3 (3.2)	2 (4.3)	0.75
EDSS progression (n, %)	13 (13.7)	5 (10.6)	0.59
EDSS improvement (n, %)	12 (12.4)	8 (17.1)	0.49
NEDA-3 (n, %)	72 (75.8)	36 (76.6)	0.92

DMT—disease modifying therapy, T2-w—T2-weighted, GELs—gadolinium enhancing lesions, EDSS—Expanded Disability Status Scale, NEDA-3—No Evidence of Disease Activity-3, * chi2 test.

**Table 3 medicina-61-01568-t003:** Adverse events after first 12-months of ofatumumab treatment between switch and naïve patients.

Adverse Events (AEs)	Patients After Switch form Another DMT (N = 95)	Naïve Patients (N = 47)	*p* *
Total (n, %)	62 (65.3)	34 (72.3)	0.40
Infusion related reactions (n, %)	57 (60.0)	31 (66.0)	0.49
Infections (n, %)	18 (18.9)	7 (14.9)	0.55

DMT—disease modifying therapy, * chi2 test.

**Table 4 medicina-61-01568-t004:** Regression analysis of factors determining clinical and radiological disease activity after first 12-months ofatumumab treatment.

	Age	Gender (Female)	Disease Duration	De Novo Treatment	EDSS at Treatment Initiation
Relapse Activity					
OR	1.02	0.69	1.01	1.09	1.28
95% CI	0.96–1.09	0.21–2.37	0.92–1.11	0.26–4.71	1.08–1.52
*p*	0.49	0.56	0.82	0.9	0.005
New or enlarged T2-w lesions					
OR	1.02	1.35	0.92	0.76	1.11
95% CI	0.96–1.07	0.39–4.66	0.82–1.03	0.22–2.58	0.93–1.31
*p*	0.55	0.63	0.14	0.67	0.24
GELs					
OR	1.07	1.03	0.99	1.15	0.88
95% CI	1.007–1.67	0.92–1.15	0.84–1.16	0.14–9.19	0.65–1.2
*p*	0.02	0.56	0.86	0.89	0.43
EDSS progression					
OR	1.02	0.7	0.99	0.89	1.11
95% CI	0.96–1.07	0.23–2.12	0.91–1.09	0.25–3.16	0.94–1.29
*p*	0.56	0.53	0.95	0.86	0.22
NEDA-3					
OR	0.99	1.21	1.01	0.98	0.91
95% CI	0.95–1.04	0.49–2.92	0.93–1.09	0.38–2.54	0.8–1.04
*p*	0.86	0.68	0.81	0.97	0.17

T2-w—T2-weighted, GELs—gadolinium enhancing lesions, EDSS—Expanded Disability Status Scale, NEDA-3—No Evidence of Disease Activity-3, OR—odds ratio, CI—confidence interval.

**Table 5 medicina-61-01568-t005:** Summary of published real-world evidence studies on ofatumumab in multiple sclerosis, including study design, patient population, follow-up, clinical and MRI outcomes, NEDA-3 rates, and safety findings.

Author, Year	Study Design, Region	N (Population)	Follow-Up	Prior DMT Exposure	Relapse Outcomes	NEDA-3	Safety/Persistence Highlights
Amin, 2025 [25]	Retrospective EMR, 2 US MS centers)	175 (80% RRMS/CIS; 15.4% SPMS; 4.6% PPMS)	12 mo	86% previously treated	1 relapse over 12 mo (~0.6%); marked drop in new T2 and GELs	NR (NEDA-2 85% in subset)	90.6% on-treatment at 12 mo; systemic IRR in 36% (mostly early); IgG stable, IgM ↓ but not linked to infections
Karl, 2024 [26]	Prospective, Germany	81 RMS	~6 mo visit (~10 mo on OFA at FU)	Mixed	4 relapses (5%); EDSS stable; no discontinuations	NR	Headache and limb pain persisted; handling rated ‘very easy’ by 88%
Stępień, 2025 [27]	Multicentre, Poland	430 RRMS	12 mo (EDSS to 24 mo)	73% previously treated; 27% naïve	Relapse-free: 45% → 88%	72.7%	AEs in 18.6% (mostly flu-like/weakness); no discontinuations; lymphocytes dipped at 2 mo then normalized
Tai, 2025 [28]	US claims, pre-post	779 MS	Mean 1.36 yrs	63% had DMT pre-index	ARR 0.41 → 0.10 (75% ↓); hospitalizations 90% ↓	NR	Safety not captured in claims
Zanghì, 2024 [29]	Italy; OFA vs. OCR, PSM; OFA arm	180 RMS	Mean 13.2 mo	Mix of naïve/switch; high-efficacy pre-DMT excluded	ARR 0.038; MRI activity 2.7%	94.4%	IRR 17.2%, URTI 11.1%, UTI 3.3%; no SAEs

ARR—annualized relapse rate; CIS—clinically isolated syndrome; CDP—confirmed disability progression; DMT—disease-modifying therapy; EDSS—Expanded Disability Status Scale; EMR—electronic medical records; FU—follow-up; GELs—gadolinium-enhancing lesions; Ig—immunoglobulin; IRR—injection-related reaction; MRI—magnetic resonance imaging; MS—multiple sclerosis; NEDA—no evidence of disease activity; NR—not reported; OCR—ocrelizumab; OFA—ofatumumab; PSM—propensity score matched; QoL—quality of life; RMS—relapsing multiple sclerosis; RRMS—relapsing-remitting multiple sclerosis; SAE—serious adverse event; SPMS—secondary progressive multiple sclerosis; PPMS—primary progressive multiple sclerosis; URTI—upper respiratory tract infection; UTI—urinary tract infection; ↓—indicates decrease/reduction compared with baseline or prior period; ~—indicates approximate value.

## Data Availability

The data presented in this study are not publicly available due to privacy restrictions.

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
