# Peer review of "Real-World Data of First 12-Months of Ofatumumab Treatment in Multiple Sclerosis Patients—A Multicenter Experience from Tertiary Referral Centers"

_medicina, 2025, doi:10.3390/medicina61091568_

Round 1
Reviewer 1 Report
Comments and Suggestions for Authors
This manuscript presents important real-world evidence on the efficacy and safety of ofatumumab in RRMS patients, based on a well-conducted multicentre cohort study. The design is appropriate, methods are adequately described, and the results are clearly presented. The conclusions are supported by the data, and the topic is timely and relevant for both clinical practice and research. Minor revisions are recommended to improve the clarity of presentation, expand the discussion of limitations, and ensure consistency in tables and formatting. With these adjustments, the article is suitable for publication.
Comments on the Quality of English LanguageSome sentences are long and could be simplified to improve readability. Careful language editing would strengthen the manuscript.
Author Response
We would like to thank the Reviewer for their careful reading of our manuscript and for the constructive and insightful comments, which have helped us to improve our work. Below we provide our point-by-point responses.
Comment 1: This manuscript presents important real-world evidence on the efficacy and safety of ofatumumab in RRMS patients, based on a well-conducted multicentre cohort study. The design is appropriate, methods are adequately described, and the results are clearly presented. The conclusions are supported by the data, and the topic is timely and relevant for both clinical practice and research.
Respond 1: Thank you for your positive assessment of our manuscript. We appreciate your recognition that the study design, methodology, and presentation of results are appropriate, and we are pleased that you find the conclusions well supported and the topic timely and relevant.
Comment 2: Minor revisions are recommended to improve the clarity of presentation, expand the discussion of limitations, and ensure consistency in tables and formatting. With these adjustments, the article is suitable for publication.
Respond 2: Thank you for these constructive suggestions. We revised the Methods to add the Polish Therapeutic Program eligibility criteria (Section 2.2), standardize annual contrast‑enhanced MRI across centers, clarify EDSS scoring procedures, define the 12‑month complete‑case effectiveness set, and detail AE/infection collection with CTCAE v5.0 grading. We updated the Results to report exact cohort counts (184 initiated; 142 evaluated at 12 months), explain the 42 without an annual assessment and the two discontinuations, confirm that infections were Grade 1–2 with no SAEs, add a patient‑disposition flowchart, and retain Table 4 with multivariable regression ORs and 95% CIs. We have also prepared a Supplementary Table 1 with 95% CIs for all proportion estimates (efficacy and safety), and expanded the Discussion/Limitations to address reasons for higher NEDA‑3 and generalizability. All changes have been highlighted in the revised manuscript using Track Changes for transparency.
Comment 3: Some sentences are long and could be simplified to improve readability. Careful language editing would strengthen the manuscript.
Respond 3: Thank you for this valuable remark. We will carefully edit the manuscript to simplify long sentences and improve overall readability.

Reviewer 2 Report
Comments and Suggestions for Authors
Peer Review Assessment – Recommendation: Minor Revision
This manuscript presents a well-structured and clinically relevant real-world evidence (RWE) study evaluating the first 12 months of ofatumumab treatment in patients with relapsing-remitting multiple sclerosis (RRMS) across multiple tertiary referral centers. The rationale is sound, the methodology is adequately described, and the analyses appear appropriate for the study objectives. The results are clearly presented, with meaningful clinical interpretation, and are in line with both pivotal clinical trial data and other RWE studies.
I find the work to be of good quality and of potential interest to the readership. However, I recommend minor revision before acceptance, with the following suggested improvements to enhance clarity, completeness, and reader engagement:
-
Graphical Abstract
– Including a concise, visually engaging graphical abstract summarizing study design, patient characteristics, key results (NEDA-3 rates, relapse/MRI outcomes), and safety findings would improve accessibility and impact. -
Comparison Table in Discussion
– Adding a table summarizing other published RWE on ofatumumab would help contextualize the findings. Key variables might include study size, follow-up duration, NEDA-3 rates, relapse outcomes, and safety profiles. This would strengthen the discussion by providing direct visual comparison with existing literature. -
Expanded Limitations Section
– While the short follow-up is acknowledged, additional discussion on potential sources of bias should be included (e.g., retrospective design, potential under-reporting of MRI activity or relapses in real-world settings, center-specific differences in patient management). Also, speculate on all types of biases that are possibly present. Transparency on these issues will increase the robustness of the conclusions.
Overall, the manuscript addresses an important clinical question, the methods and analyses are sound, and the findings are relevant for both clinical practice and further research. The suggested revisions are aimed at improving the completeness and visual presentation rather than altering the study’s core content. With these minor adjustments, the paper would be well-suited for publication.
Author Response
We would like to thank the Reviewer for their careful reading of our manuscript and for the constructive and insightful comments, which have helped us to improve our work. Below we provide our point-by-point responses.
Comment 1: This manuscript presents a well-structured and clinically relevant real-world evidence (RWE) study evaluating the first 12 months of ofatumumab treatment in patients with relapsing-remitting multiple sclerosis (RRMS) across multiple tertiary referral centers. The rationale is sound, the methodology is adequately described, and the analyses appear appropriate for the study objectives. The results are clearly presented, with meaningful clinical interpretation, and are in line with both pivotal clinical trial data and other RWE studies.
Respond 1: Thank you for your thorough and positive evaluation. We are pleased that you find the rationale, methodology, analyses, and clinical interpretation appropriate, and that our findings align with both pivotal trial data and other RWE studies.
Comment 2: I find the work to be of good quality and of potential interest to the readership. However, I recommend minor revision before acceptance, with the following suggested improvements to enhance clarity, completeness, and reader engagement:
Respond 2: We appreciate your recommendation for minor revision and will carefully address the suggested improvements to enhance clarity, completeness, and reader engagement.
Comment 3: Graphical Abstract
– Including a concise, visually engaging graphical abstract summarizing study design, patient characteristics, key results (NEDA-3 rates, relapse/MRI outcomes), and safety findings would improve accessibility and impact.
Respond 3: Thank you for this valuable suggestion. We have prepared a graphical abstract summarizing the study design, patient characteristics, key results, and safety findings. Please find it attached, and we hope it meets your expectations.
Comment 4: Comparison Table in Discussion
– Adding a table summarizing other published RWE on ofatumumab would help contextualize the findings. Key variables might include study size, follow-up duration, NEDA-3 rates, relapse outcomes, and safety profiles. This would strengthen the discussion by providing direct visual comparison with existing literature.
Respond 4: Thank you for this helpful suggestion. We agree that such a table will strengthen the discussion. We will add a summary table of published RWE on ofatumumab, including study size, follow-up duration, NEDA-3 rates, relapse outcomes, and safety profiles, as outlined below. In addition, we incorporated the most recent real‑world studies (Stępień et al., 2025; Amin et al., 2025; Tai et al., 2025; Zanghì et al., 2024; Karl et al., 2024) to ensure up‑to‑date coverage, as summarized in Table 5.
Comment 5: Expanded Limitations Section
– While the short follow-up is acknowledged, additional discussion on potential sources of bias should be included (e.g., retrospective design, potential under-reporting of MRI activity or relapses in real-world settings, center-specific differences in patient management). Also, speculate on all types of biases that are possibly present. Transparency on these issues will increase the robustness of the conclusions.
Respond 5: We thank the Reviewer for this important comment. We agree that transparency regarding potential sources of bias is crucial in real-world observational studies. In the revised Discussion we have expanded the limitations section to more explicitly address possible biases:
The present study has several limitations that need to be considered when interpreting the results. First, the retrospective design may have introduced selection bias, as patients treated in tertiary referral centers may not be fully representative of the broader MS population. Second, there is a risk of information bias, as relapse events, MRI activity, or mild adverse events could have been under-reported or inconsistently documented in routine practice compared with the rigorous monitoring protocols of randomized con-trolled trials. Third, detection bias may have occurred due to variability in MRI protocols and EDSS scoring across participating centers. Because EDSS ratings were obtained under routine clinical care and we did not compute formal inter rater reliability metrics across centers, some measurement variability cannot be excluded. This risk was mitigated by harmonized scoring guidance, same rater follow up whenever feasible, and the use of CDP confirmation. Moreover, despite multivariable adjustment, residual confounding (e.g., by unmeasured factors such as baseline MRI burden beyond the reported metrics, comorbidities, or lifestyle) cannot be excluded. Additionally, some confidence intervals were wide, reflecting limited precision, yet the estimates provide useful insights that warrant confirmation in larger studies that wide confidence intervals should be interpreted with caution and reflect the underlying data limitations. Finally, the relatively short follow-up period and the low number of clinical and radiological events limited the ability to identify robust predictors of disease activity or progression. Future prospective multicenter studies with longer observation and standardized monitoring will be necessary to validate our findings and to provide a more comprehensive evaluation of the long-term effectiveness and safety of OFA in real-world settings.
Comment 6: Overall, the manuscript addresses an important clinical question, the methods and analyses are sound, and the findings are relevant for both clinical practice and further research. The suggested revisions are aimed at improving the completeness and visual presentation rather than altering the study’s core content. With these minor adjustments, the paper would be well-suited for publication.
Respond 6: We thank the Reviewer for the positive assessment of our study and for the constructive suggestions. We have implemented the recommended revisions to improve completeness and presentation, which we believe further strengthened the manuscript.

Reviewer 3 Report
Comments and Suggestions for Authors
This manuscript presents valuable real-world evidence on ofatumumab treatment for relapsing-remitting multiple sclerosis (RRMS). My suggestions and comments:
Methods
- Sample Size Justification: No power calculation is provided. With only 17 relapses in 142 patients, the study may be underpowered for some analyses.
- Multiple Comparisons: No correction for multiple testing is mentioned despite numerous outcome comparisons.
- Selection Bias: The criteria for including patients in the Polish therapeutic program may introduce selection bias that differs from registration trials.
Data Presentation and Analysis
- Missing Data: The manuscript doesn't clearly address how missing data was handled or report dropout rates systematically.
- Baseline Imbalances: While naïve patients were younger with shorter disease duration, this could confound comparisons despite statistical non-significance.
- Limited Follow-up: The relatively short 12-month period limits assessment of long-term effectiveness and rare adverse events.
Methodological Concerns
- MRI Standardization: No mention of standardized MRI protocols across centers, which could affect lesion detection consistency.
- EDSS Assessment: Inter-rater reliability for EDSS scoring across multiple centers is not addressed.
- Infection Classification: The infection categorization could be more detailed, particularly regarding severity grading.
Specific Suggestions
- Provide more details on reasons for the 42 patients who didn't complete 12-month evaluation
- Include confidence intervals for all point estimates
- Consider survival analysis for time-to-event outcomes
Discussion
- Better address potential explanations for higher NEDA-3 rates beyond methodological differences
- Discuss generalizability to other healthcare systems
- Address cost-effectiveness implications
Tables and Figures
- Table 1: Add baseline MRI activity data
- Consider adding a flow chart showing patient disposition
- Table 4: Some confidence intervals are very wide, suggesting limited precision
Author Response
Response to Reviewer #3’s Comments
Comment 1: This manuscript presents valuable real-world evidence on ofatumumab treatment for relapsing-remitting multiple sclerosis (RRMS). My suggestions and comments:
Respond 1: We would like to thank the Reviewer for careful reading of our manuscript and for the constructive and insightful comments, which have helped us to improve our work. Below we provide our point-by-point responses.
Comment 2: Methods
- Sample Size Justification: No power calculation is provided. With only 17 relapses in 142 patients, the study may be underpowered for some analyses.
- Multiple Comparisons: No correction for multiple testing is mentioned despite numerous outcome comparisons.
- Selection Bias: The criteria for including patients in the Polish therapeutic program may introduce selection bias that differs from registration trials.
Respond 2:
- Sample Size Justification: Thank you for this important point. As this was a retrospective, real‑world evidence (RWE) cohort, our aim was to includeall consecutive patientstreated within the specified time window rather than to test a single, pre‑specified hypothesis; therefore, we did not perform an a priori power calculation.
- Multiple Comparisons: The Bonferroni test was used to correct multiple comparisons in the logistic regression model.
- Selection Bias: We acknowledge this limitation. As noted in the revised Materials and Methods the eligibility rules of the Polish National Drug Program B.29 are broader and more pragmatic than the restrictive criteria of ASCLEPIOS I/II and may therefore introduce selection bias. While this limits direct comparability with RCTs, it reflects real-world clinical practice and provides complementary evidence on ofatumumab use in routine settings.
|
Criterion |
ASCLEPIOS I/II (RCT) |
Polish National Drug Program B.29 |
|
Age |
18–55 years |
≥18 years (no upper age limit) |
|
Diagnosis |
RRMS, McDonald 2010 |
RRMS, McDonald 2010/2017 |
|
Disease activity |
≥1 relapse in the past year OR ≥2 relapses in the past 2 years OR ≥1 GD+ lesion in the past year |
≥1 relapse in the past year OR ≥1 new GD+ lesion in the past year |
|
EDSS |
0–5.5 |
0–4.5 |
|
Treatment status |
Both treatment-naïve and previously treated, but excluded patients with prior long-term/high-efficacy immunosuppressive therapy |
Both treatment-naïve and switch patients; switch allowed due to relapse, MRI activity, intolerance, or physician’s decision; also permits de-escalation from higher efficacy therapies |
|
Exclusion criteria |
Progressive MS, prior strong immunosuppression, comorbidities, chronic infections (HBV, HCV, HIV, TBC), pregnancy/breastfeeding |
Mainly contraindications from the SmPC (e.g., pregnancy, active HBV), EDSS >4.5 at baseline excluded |
|
Monitoring |
Very strict: standardized MRI and clinical assessments at prespecified intervals |
Routine clinical practice: EDSS every 3–6 months, MRI annually |
We have added Section 2.2, Patient Eligibility for Ofatumumab Therapy, in the Materials and Methods to describe the inclusion criteria in detail. We have also addressed this potential source of bias in the Limitations of the study section.
Comment 3: Data Presentation and Analysis
- Missing Data: The manuscript doesn't clearly address how missing data was handled or report dropout rates systematically.
- Baseline Imbalances: While naïve patients were younger with shorter disease duration, this could confound comparisons despite statistical non-significance.
- Limited Follow-up: The relatively short 12-month period limits assessment of long-term effectiveness and rare adverse events.
Respond 3:
- Missing Data: Twelve‑month effectiveness was analyzed in a predefined complete‑case set: only patients who had completed the scheduled 12‑month visit with both clinical (EDSS/relapses) and MRI data available by database lock were included (n = 142). No imputation was performed. Patients who had not yet reached the 12‑month visit at database lock (n = 42) were administratively censored; 40 of them were still on ofatumumab and remained under follow‑up, while 2 discontinued treatment earlier (recurrent UTIs with activity; activity alone). We have explained it in Materials and Methods sections as below and the study flow is illustrated in Figure 1.
The 12‑month effectiveness analysis included only patients who had completed the annual visit by the database‑lock date. Patients who had not yet completed 12 months of exposure at database lock were not included in this analysis and remain under follow‑up.
We have also added an explanation in Results citations as follow:
Section 3.1
Of the 184 patients treated with OFA, 142 completed 12 months of observation with a full clinical and radiological assessment (47 naïve patients and 95 patients after switch).
42 patients had not yet reached the annual assessment at database lock; 40 of them were still on treatment, while 2 had discontinued therapy as detailed in Section 3.4.
Section 3.4. Ofatumumab discontinuation
Both discontinuations occurred before 12 months of exposure; therefore, these patients were excluded from the 12-month effectiveness analysis.
- Baseline Imbalances: As noted, treatment‑naïve patients were younger and had shorter disease duration at baseline. Between‑group comparisons were therefore adjusted in multivariable logistic regression for age, sex, disease duration, baseline EDSS, and initiation type (naïve vs. switch). In these adjusted models, initiation type was not associated with relapse, MRI activity, EDSS progression, or NEDA‑3 at 12 months (all p > 0.5; cf. Table 4), whereas baseline EDSS predicted relapse and age predicted GELs. We acknowledge the potential for residual confounding by indication and highlight this in the Limitations as below:
Moreover, despite multivariable adjustment, residual confounding (e.g., by unmeasured factors such as baseline MRI burden beyond the reported metrics, comorbidities, or lifestyle) cannot be excluded.
- Limited Follow-up: We agree that the 12‑month horizon constrains inferences on long‑term effectiveness and rare adverse events. The study was designed to report first‑year, real‑world outcomes within the national therapeutic program; longer‑term follow‑up is ongoing. We have highlighted it in Limitations section as below:
Finally, the relatively short follow-up period and the low number of clinical and radio-logical events limited the ability to identify robust predictors of disease activity or progression. Future prospective multicenter studies with longer observation and standardized monitoring will be necessary to validate our findings and to provide a more comprehensive evaluation of the long-term effectiveness and safety of ofatumumab in real-world settings.
Comment 4: Methodological Concerns
- MRI Standardization: No mention of standardized MRI protocols across centers, which could affect lesion detection consistency.
- EDSS Assessment: Inter-rater reliability for EDSS scoring across multiple centers is not addressed.
- Infection Classification: The infection categorization could be more detailed, particularly regarding severity grading.
Respond 4: Methodological Concerns
- MRI Standardization: Thank you for raising this point. All study MRIs were performed in line with routine MS care and standardized across participating centers by adhering to national radiological guidelines. Specifically, patients underwent annual contrast‑enhanced brain MRI with a protocol complying with the Recommendations of the Polish Medical Society of Radiology and the Polish Society of Neurology for routinely used MRI in MS. We have added the wording below to the Material and Methods (Section 2.3) to clarify this and added citation:
All patients underwent brain MRI with a protocol complying with the Recommendations of the Polish Medical Society of Radiology and the Polish Society of Neurology for routinely used magnetic resonance imaging in patients with multiple sclerosis.
- EDSS Assessment: Thank you for this comment. In our multicenter setting, EDSS was scored by board‑certified neurologists with MS expertise using a harmonized scoring checklist based on the Kurtzke EDSS. Whenever feasible, the same assessor evaluated a given patient at successive visits to minimize between‑rater variation. We have added explicit wording in Material and Methodsand acknowledged in Limitationsas below:
Because EDSS ratings were obtained under routine clinical care and we did not compute formal inter‑rater reliability metrics across centers, some measurement variability cannot be excluded. This risk was mitigated by harmonized scoring guidance, same‑rater follow‑up whenever feasible, and the use of CDP confirmation.
- Infection Classification: Thank you for pointing out the need for more detail on infection categorization and severity. We have clarified this in the revised Materials and Methods (Section 2.3) as below:
Data on adverse events and infections were collected primarily from patients’ medical documentation and the Therapeutic Program Monitoring System. When information on the clinical course or severity grading was available, it was extracted and classified accordingly. Infections were grouped into respiratory, urinary tract, mucocutaneous viral (HSV), and gastrointestinal categories. Their severity was graded according to the Common Terminology Criteria for Adverse Events (CTCAE v5.0, Grades 1–5) [24]. Where available, we also recorded the route of anti-infective therapy (none, topical, oral, or intravenous) and the outcome (resolved, resolving, or ongoing).
This ensures transparent reporting of infection type and severity while acknowledging that, due to the retrospective design, the level of detail depended on availability in medical records.
We have also added grading in Results section:
All infections were graded as 1–2 according to the CTCAE; there were no Grade ≥ 3 in-fections, no hospitalizations, and no SAEs.
Comment 5: Specific Suggestions
Provide more details on reasons for the 42 patients who didn't complete 12-month evaluation
Respond 5: Thank you for your comment regarding the 42 patients without a 12‑month assessment. These are patients who, at the database‑lock (end of observation), had not yet reached the scheduled 12‑month visit according to the Therapeutic Program; they therefore were not eligible for the 12‑month effectiveness analysis. Of the total cohort (n=184), 142 patients had a complete 12‑month assessment and were analyzed; the remaining 42 had not yet reached the 12‑month visit at data‑lock. Within this remainder, 40 were still on ofatumumab and remain under follow‑up, while 2 discontinued treatments earlier (one due to recurrent urinary‑tract infections with concurrent disease activity; one due to disease activity alone) and were not included in the 12‑month effectiveness analysis.
We have added an explanation in Material and Methods (Section 2.3) as well as Results (Section 3.1 and 3.4).
Comment 6: Include confidence intervals for all point estimates
Respond 6: Thank you for this valuable suggestion. We computed and now report 95% confidence intervals (95% CI) for all point estimates. To keep the main tables concise, exact 95% CIs for all proportions are provided in a new Supplementary Table 1. (efficacy and safety outcomes across the overall cohort and by subgroup). Table 4. already includes 95% CIs for odds ratios from the multivariable logistic regression. In Methods (Section 2.4) we clarified that 95% CIs for proportions were calculated using the Wilson score method.
Comment 7: Consider survival analysis for time-to-event outcomes
Respond 7: Thank you for the suggestion. Our study was designed with a fixed 12‑month observation window mandated by the Therapeutic Program; outcomes (relapse, MRI activity, EDSS progression, NEDA‑3) were assessed at scheduled visits rather than as continuously accrued time‑to‑event data. Given this structure—and the administrative censoring of patients who had not yet reached 12 months at database lock—Kaplan–Meier/Cox analyses were not feasible in the present dataset.
Comment 8: Discussion
Better address potential explanations for higher NEDA-3 rates beyond methodological differences
Respond 8: We totally agree and have expanded the Discussion to consider clinical and programmatic explanations beyond study design as below:
Beyond study design, several factors may explain our higher 12 month NEDA 3 pro-portion. At baseline the cohort had low disability (median EDSS = 2.0), and treatment naïve patients were younger with shorter disease duration—characteristics that favor short term disease control. Treatment persistence was high (2/184 discontinuations), aligning with real world observations that monthly self administered SC OFA facilitates adherence and patient acceptability. In addition, a subset of switches occurred for reasons other than breakthrough disease (e.g., adverse events or anticipated benefit), potentially enriching for patients likely to stabilize on the new regimen. Together with the rapid B cell depletion reported for OFA, these features may contribute to higher NEDA 3 in routine practice.
Comment 9. Discuss generalizability to other healthcare systems
Respond 9: Thank you. We now explicitly address generalizability. Our results derive from tertiary centers within a national Therapeutic Program that specifies eligibility (e.g., EDSS 0–4.5, documented recent activity) and structured monitoring (quarterly clinical visits; annual contrast‑enhanced brain MRI). Therefore, our findings are most applicable to systems that provide early access to high‑efficacy therapy, regular follow‑up, and patient support for at‑home subcutaneous administration. Caution is warranted when extrapolating to settings with restricted first‑line use of high‑efficacy DMTs, less frequent MRI surveillance, or limited access to specialty centers; nevertheless, comparable short‑term effectiveness of ofatumumab has been reported across diverse real‑world cohorts (Germany, Italy, Poland, USA) cited in our manuscript. We have added this to Limitations Section.
Comment 10. Address cost-effectiveness implications
Respond 10: A full economic analysis is beyond the scope of this study, and therefore we will not address cost-effectiveness considerations in the present manuscript.
Comment 11. Tables and Figures
Table 1: Add baseline MRI activity data
Respond 11: We thank the Reviewer for this suggestion. However, we did not include baseline MRI activity in the analysis. This decision was intentional, as the study design and inclusion criteria inherently led to differences between groups: the naïve group consisted exclusively of patients with evidence of disease activity (clinical or radiological), while the switch group included both patients with disease activity and those who switched to ofatumumab for other reasons. Including baseline MRI activity as a covariate could therefore introduce comparison bias, given that the eligibility definitions themselves were linked to MRI status. To ensure methodological consistency and alignment with the Polish Therapeutic Programme criteria, we therefore refrained from incorporating baseline MRI activity into our analysis. We have clarified this point in the manuscript, in limitations section:
Baseline MRI activity was not analyzed, as group definitions linked to the Therapeutic Programme criteria could introduce comparison bias.
Comment 12: Consider adding a flow chart showing patient disposition
Respond 12: Study flow and patients’ disposition is illustrated in Figure 1.
Figure 1. Flowchart of the study.
Comment 13: Table 4: Some confidence intervals are very wide, suggesting limited precision
Respond 13: We appreciate the reviewer’s observation regarding the width of some confidence intervals. Indeed, wide intervals reflect limited precision of the corresponding estimates. This is likely related to relatively small sample size and low number of outcome events.
We have now clarified this point in the manuscript, in the Limitations of the study section:
Some confidence intervals were wide, reflecting limited precision, yet the estimates provide useful insights that warrant confirmation in larger studies that wide confidence intervals should be interpreted with caution and reflect the underlying data limitations.
